# Beyond Canonicalization: How Tensorial Messages Improve Equivariant Message Passing

**Peter Lippmann**[*]**, Gerrit Gerhartz**[*]**, Roman Remme & Fred A. Hamprecht**
Interdisciplinary Center for Scientific Computing (IWR), Heidelberg University,
69120 Heidelberg, Germany
{peter.lippmann,roman.remme,fred.hamprecht}@iwr.uni-heidelberg.de
gerrit.gerhartz@proton.me

## Abstract

In numerous applications of geometric deep learning, the studied systems exhibit spatial symmetries and it is desirable to enforce these. For the symmetry of global rotations and reflections, this means that the model should be equivariant with respect to the transformations that form the group of $O(d)$. While many approaches for equivariant message passing require specialized architectures, including non-standard normalization layers or non-linearities, we here present a framework based on local reference frames ("local canonicalization") which can be integrated with any architecture without restrictions. We enhance equivariant message passing based on local canonicalization by introducing tensorial messages to communicate geometric information consistently between different local coordinate frames. Our framework applies to message passing on geometric data in Euclidean spaces of arbitrary dimension. We explicitly show how our approach can be adapted to make a popular existing point cloud architecture equivariant. We demonstrate the superiority of tensorial messages and achieve state-of-the-art results on normal vector regression and competitive results on other standard 3D point cloud tasks.

## 1 Introduction

Data from various domains, such as scans of 3D scenes, molecules or earth science data, consists of a set of nodes positioned in Euclidean space and equipped with geometric node features. In many tasks, message passing neural networks are used to extract and combine these node features. In application domains in which inputs and outputs are governed by known symmetries, it may be desirable or required to enforce these. One such approach is to build equivariant architectures, which guarantee that the learned function behaves in a well-defined manner under transformations of the input. Regarding rotations and reflections specifically, equivariance ensures that predictions are consistent for different orientations of the input. The idea of equivariance is not only conceptually appealing but also highly relevant in practice: Built-in equivariance is known to enhance performance and data efficiency of neural networks in several settings (Weiler et al., 2018; Batzner et al., 2022). For instance, in deep learning based molecular dynamics simulations, exact equivariance can be crucial for the simulation stability (Fu et al., 2022). However, exact equivariance is often realized by restricting the model design to specialized building blocks such as non-standard linear layers, normalization layers and non-linearities. Some of these specialized building blocks are computationally intensive Passaro & Zitnick (2023). Alternative approaches, which do not require specialized building blocks, are typically based on reference frames. The reference frames are used to transform geometric inputs into a canonical orientation before feeding them to the model ("canonicalization"). In many geometric learning tasks, the model inputs are comprised of geometric substructures, in which case local reference frames seem most suitable to orient each substructure individually into a canonical pose ("local canonicalization"). However, communicating geometric information consistently between local patches with *different* coordinate frames is non-trivial (cf. Fig. 1). In this paper,

---
[*]equal contribution

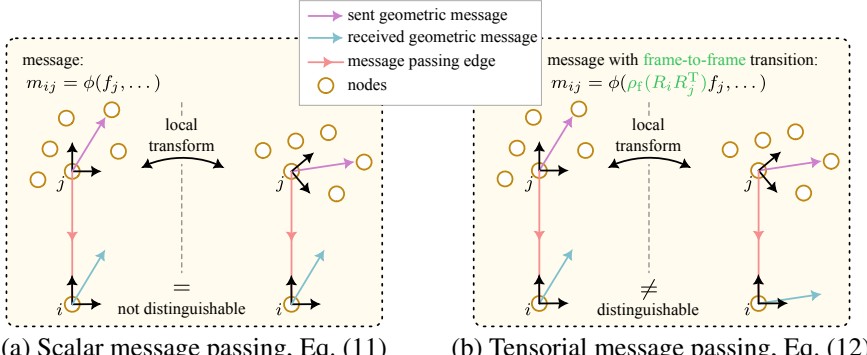

(a) Scalar message passing, Eq. (11)  (b) Tensorial message passing, Eq. (12)

Figure 1: **Limitation of scalar message passing.** (a) The upper node $j$ is sending a characteristic direction in its neighborhood (encoded in a vector) to an adjacent node $i$. If the local neighborhood of $j$ is rotated, the vector and the equivariant local frame rotate along. The coordinates of the vector are invariant and thus, with scalar message passing, node $i$ will receive the same message, despite the two geometries being different. (b) Tensorial message passing overcomes this limitation so that directional information can be sent consistently.

we present a novel framework that provides a practical solution to this problem and allows building more expressive equivariant architectures based on local canonicalization. Concretely, we make the following contributions:

- We present a novel message passing formalism that together with local canonicalization enables consistent communication of geometric features and can be used to build $O(d)$-equivariant message passing networks.

- We explicitly show how to adapt our framework to make any existing message passing architecture $O(d)$-equivariant. As a concrete example, we present an $O(3)$-equivariant adaptation of the widely-used PointNet++ architecture (Qi et al., 2017b), which produces state-of-the-art results.

- We demonstrate conceptually and experimentally that tensorial messages are a strict generalization of vanilla local canonicalization methods and that tensorial messages improve the performance.

- We propose to update the local frames after each layer of message passing so that geometric information aggregated from the neighborhood of each node can be used to iteratively refine the local frames. This is a strict generalization of static local canonicalization.

- Our framework allows for a direct comparison between equivariant architectures and non-equivariant ones. We find that exact, built-in equivariance via our framework is more data-efficient and outperforms our strong baseline models trained via data augmentation.

## 2 RELATED WORK

Many popular architectures for message passing on point cloud data are not equivariant per se (Qi et al., 2017b; Liu et al., 2019; Wang et al., 2019). In general, the simplest way to achieve (approximate) equivariance with respect to a set of transformations is to augment the training data, i.e. present the model with randomly transformed samples during training. Data augmentation is of course completely independent of the model and does not constrain the architecture. However, the equivariance must be learned and is thus not guaranteed, meaning that it may not generalize to out-of-distribution samples; and the learned equivariance is not exact. For a comprehensive overview of different approaches to equivariant message passing we refer the reader to (Duval et al., 2023).

**Equivariance using tensorial internal representations.** Invariance is a special case of equivariance where the output stays invariant when the input is transformed. A simple way of achieving exact invariance is to extract invariant features from the input geometry, such as distances or angles, and only include these in the message passing (Schütt et al., 2018; Li et al., 2021b). While

this allows using typical deep learning building blocks (linear layers, activations, norm layers, etc.), these approaches are not able to communicate non-scalar geometric information (such as directions) during message passing. Thus, several works have included vectorial and tensorial features into the message passing formalism to predict equivariant quantities (Deng et al., 2021; Satorras et al., 2021; Frank et al., 2022; Batatia et al., 2022; Liao et al., 2024; Musaelian et al., 2023; Remme et al., 2023; Simeon & De Fabritiis, 2024). It has been shown that including non-scalar geometric features in the message passing can enhance the performance, even when the model outputs are invariant quantities (Fuchs et al., 2020; Brandstetter et al., 2022) and that higher-order tensor representations are particularly helpful in tasks where angular information matters, e.g. for predicting forces in molecules (Zitnick et al., 2022; Batzner et al., 2022). However, these architectures may no longer treat every internal activation as individual number, for instance, the coordinates of vectors must be processed jointly to maintain equivariance. In contrast to our framework, these approaches therefore rely on carefully designed non-linearities, norm layers and special operators to combine scalar and tensorial features, see e.g. (Thomas et al., 2018) for details.

**Equivariance by canonicalization.** An alternative approach to achieve invariance is to determine a canonical *global* orientation of the input point cloud in the first layers and then canonicalize the input accordingly before feeding it into the main model. This factors out the global orientation of the input so that the model output will be invariant. Several methods have been developed to predict the global orientation, based on subnetworks (Qi et al., 2017a), principal component analysis (Li et al., 2021a), asymmetric units (Baker et al., 2024), as a combination of local orientations (Zhao et al., 2020; 2022) or based on anchor points (Lou et al., 2023). Alternatively, Puny et al. (2021) propose to use several frames and average their predictions to achieve equivariant predictions based on non-equivariant backbone architectures.

Similar to equivariance by global orientation estimation, one can achieve equivariance using *local* canonicalization (Zhang et al., 2020; Wang & Zhang, 2022; Kaba et al., 2023). As in our approach, one equivariantly predicts a local coordinate frame for each node, into which the geometric input features are transformed (Luo et al., 2022; Du et al., 2022; 2024). Thereby, the local coordinates of the node features are independent of the global orientation of the input and are thus invariant. One advantage of using local reference frames over a single global one is that substructures in the data, which are geometrically similar, are described by similar local features. During message passing, most previous works do not leverage the local coordinate frames to transform features between the frames, which substantially limits the communication between nodes that have different local frames (cf. Fig. 1). While (Wang & Zhang, 2022) and (Du et al., 2024) try to learn an approximate frame-to-frame transition, no previous work uses general geometric representations to directly transform geometric features from one local frame into the other during message passing. For mesh CNNs, the approach of (Cohen et al., 2019) introduces transformations between local frames via parallel transport. Their approach uses $\mathrm{SO}(2)$ gauge equivariant kernels to obtain predictions which are independent of the choice of gauge. In contrast to our work, gauge equivariant approaches constrain the convolution kernels and require specialized non-linearties. In our framework, computations explicitly depend on the local coordinate frames and we demonstrate experimentally that, for our approach, informative local frames can improve model performance.

## 3 PRELIMINARIES

The set of transformations associated with symmetries of the data typically forms a group in the mathematical sense. Therefore, we will formally define group representations, which characterize the well-defined frame-to-frame transformation behavior of node features in our message passing framework.

**Group representation.** Given a group $G$ and a vector space $V$, a *group representation* $\rho$ is a mapping from $G$ to the invertible matrices $\mathrm{GL}(V)$ that fulfills

$$\rho(g_1 g_2) = \rho(g_1)\rho(g_2), \quad \forall g_1, g_2 \in G, \tag{1}$$

where $g_1 g_2$ is the group product and $\rho(g_1)\rho(g_2)$ the matrix product. The representation specifies how elements of the group act on vectors $v \in V$, i.e. in components $(\rho(g)v)_i = \rho(g)_{ij}v_j$. We are consistently using the Einstein summation convention throughout this paper, meaning that indices

Figure 2: **Expressive $O(d)$-equivariant message passing based on local canonicalization with tensorial messages.** Based on the input geometry, one predicts an equivariant local frame $R_i$ at each node $i$. The geometric input node features $F_i$ are transformed from the global reference frame into the respective local frames, yielding coordinates $f_i$ invariant to the choice of global frame. In order to communicate geometric information consistently during message passing, node features are treated as vectors and tensors which are transformed from one local frame into another. After each message passing layer, the local frames are refined to incorporate newly aggregated geometric information. Finally, the geometric node features are transformed back from the local into the global frame to produce an equivariant output.

that appear twice are summed over. The dimension of the representation $\rho$ is defined to be the dimension of $V$. Condition (1) implies that $\rho(g^{-1}) = (\rho(g))^{-1}$.

**Equivariance.** Let $G$ be a group and $V, W$ two vector spaces equipped with group representations $\rho_{\text{in}}$ and $\rho_{\text{out}}$ respectively. A function $\varphi : V \to W$ is said to be *equivariant* under the group $G$ if the following holds:

$$\rho_{\text{out}}(g)\varphi(x) = \varphi(\rho_{\text{in}}(g)x), \quad \forall g \in G, \, \forall x \in V$$

where the input to $\varphi$ transforms under the representation $\rho_{\text{in}} : V \to \text{GL}(V)$ and its output under the representation $\rho_{\text{out}} : W \to \text{GL}(W)$. If $\rho_{\text{out}}(g) = id$ for all $g \in G$, the function $\varphi$ is said to be *invariant*.

**Tensor representation.** Let us consider the group of rotations and reflections $O(d)$ that consists of $d \times d$ orthogonal matrices. A $d$-dimensional vector $v$ transforms under $R \in O(d)$ by contraction of its only index, i.e. $(Rv)_i = R_{ij}v_j$, forming a representation according to the above definition. Higher-order *tensor representations* are formed according to the following transformation rule:

$$T'_{i_1 \ldots i_n} = R_{i_1 j_1} \ldots R_{i_n j_n} T_{j_1 \ldots j_n}. \tag{2}$$

One may easily check that this transformation behavior fulfills condition (1) and defines a representation (see App. A). A tensor $T_{j_1 \ldots j_n}$ with $n$ indices is said to have order $n$. All indices run over $d$ dimensions. Consequently, the vector space on which this representation acts is $d^n$-dimensional.

The fact that the orthogonal matrices also include reflections can be used to distinguish the transformation behavior of geometric objects with respect to orientation-reversing transformations (with determinant $-1$). Since the determinant is multiplicative, the following transformation behavior defines another representation, namely the one for *pseudotensors*:

$$P'_{i_1 \ldots i_n} = \det(R) R_{i_1, j_1} \ldots R_{i_n, j_n} P_{j_1 \ldots j_n}. \tag{3}$$

For instance, a 3D pseudovector $\mathbf{v}$ does not change sign under parity, e.g. $\rho(P)\mathbf{v} = \mathbf{v}$ for $P = \text{diag}(-1, -1, -1) \in O(3)$, see e.g. (Jeevanjee, 2011) for mathematical details.

**Local and global frames.** The global (reference) frame is the coordinate system in which the coordinates of geometric inputs and outputs are expressed. It agrees across all nodes. Local (reference) frames are local coordinate systems represented by an $O(d)$ matrix that transforms a $d$-dimensional vector from the global frame into the corresponding local frame. We use an individual local frame for each node.

## 4 METHODS

The central idea behind our framework (Fig. 2) can be summarized as follows: for every node, one predicts an orthonormal local frame in an equivariant fashion, meaning that it transforms consis-

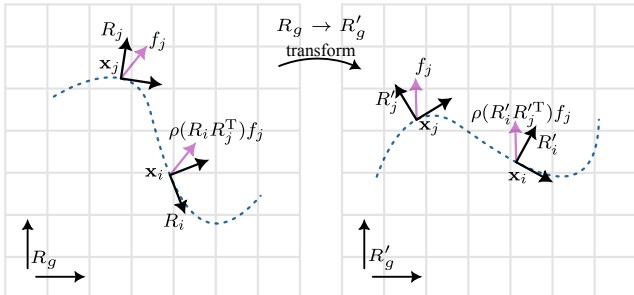

Figure 3: **Illustration of tensorial message passing between local frames.** Node $j$ sends a vectorial feature $f_j$ from its local frame $R_j$ to node $i$ with local frame $R_i$. Through the change of basis, the vectorial information can be received in the other coordinate frame without loss of information. Due to the equivariance of the local frames, the local frame coordinates of geometric objects are invariant under global transformation.

tently as the input point cloud is flipped or rotated. Then, one expresses the geometric node features in the respective local frame by a change of basis from the global reference frame to the local one (known as "local canonicalization"). Crucially, since the local frames are chosen equivariantly, the node features expressed in the local frames are invariant under $\mathrm{O}(d)$-transformations of the input (see Fig. 3). Below, we provide a proof that this indeed holds for all geometric objects, irrespective of their representation. The invariant coordinates can then be processed by arbitrary functions without breaking the invariance. The key ingredient in our approach is the following: During message passing, node features are transformed from one local frame into another. This enables direct communication of geometric features, like vectors and tensors, resulting in a strictly more expressive message passing formulation. Without the change of basis geometric information may be lost (cf. Fig. 1). At the final layer, one transforms the invariant numbers back into a geometric object in the global frame, using the desired output representation. One thereby obtains an equivariant prediction.

## 4.1 EQUIVARIANCE BY LOCAL CANONICALIZATION

Let us first describe how to predict the local frames in an equivariant manner, and illustrate how they are used to construct an equivariant pipeline. For concreteness, we will describe the procedure in three-dimensional Euclidean space, though all statements generalize straightforwardly to higher dimensions.

Geometrically informative local frames should be constructed robustly, i.e. small changes in the local geometry should not change them drastically. Secondly, they should be predicted equivariantly, i.e. if the input is transformed, the local frames must transform accordingly. We adapt the simple approach by (Wang & Zhang, 2022) to learn equivariant local frames during training. Given an input graph of nodes with coordinates $\mathbf{x}_i$ and input node features $F_i$, we equivariantly predict two vectors $\mathbf{v}_{i,1}$ and $\mathbf{v}_{i,2}$ for each node $i$:

$$\mathbf{v}_{i,k} = \sum_{j \in \mathcal{N}(i)} \left[ \omega(\|\mathbf{x}_i - \mathbf{x}_j\|) \, \phi(F_i^{(s)}, F_j^{(s)}, e_{ij}^{(s)}, \|\mathbf{x}_i - \mathbf{x}_j\|)_k \, \frac{\mathbf{x}_i - \mathbf{x}_j}{\|\mathbf{x}_i - \mathbf{x}_j\|} \right], \quad k \in \{1, 2\}, \quad (4)$$

where $\|\cdot\|$ denotes the Euclidean norm and $\mathcal{N}(i)$ the neighborhood of node $i$. For point cloud data without any edges, the neighborhood is obtained from a radius graph with cutoff radius $r_c$. $\phi$ is a standard MLP that receives the even scalars (invariant under rotations and reflections) among the node features $F_i^{(s)}, F_j^{(s)}$ and edge attributes $e_{ij}^{(s)}$ (if available) as inputs. The vectors $\mathbf{v}_{i,1}$ and $\mathbf{v}_{i,2}$ are computed as weighted sums of the normalized edge vectors, with the two outputs of $\phi$ being the respective weights. $\omega$ is an envelope function adapted from (Gasteiger et al., 2020), which goes to zero smoothly at the cutoff radius $r_c$ (see App. B for details). Using the Gram-Schmidt procedure, one equivariantly constructs two orthonormal vectors from $\mathbf{v}_{i,1}, \mathbf{v}_{i,2}$ (see App. B). A third vector is obtained from the vector product between these two, yielding an orthonormal basis $\mathbf{n}_{i,k} \in \mathbb{R}^3$, $k = 1, 2, 3$. However, the vector product $\mathbf{n}_{i,3} = \mathbf{n}_{i,1} \times \mathbf{n}_{i,2}$ always results in a right-

handed local frame so that the handedness would not change under reflection of the input. Hence, such local frames would be SO(3)- but not O(3)-equivariant. A simple solution is to flip the third vector based on the local center of mass $\bar{\mathbf{r}}$:

$$\mathbf{n}_{i,3} = \begin{cases} \mathbf{n}_{i,1} \times \mathbf{n}_{i,2} & \text{if} \quad (\mathbf{n}_{i,1} \times \mathbf{n}_{i,2}) \cdot \bar{\mathbf{r}} > 0 \\ -\mathbf{n}_{i,1} \times \mathbf{n}_{i,2} & \text{else} \end{cases} \quad , \text{ with } \bar{\mathbf{r}} := \sum_{j \in \mathcal{N}(i)} \omega(\|\mathbf{x}_j - \mathbf{x}_i\|)(\mathbf{x}_j - \mathbf{x}_i),$$

(5)

where $\cdot$ denotes the standard dot-product. The computation of $\bar{\mathbf{r}}$ is smoothed using the same envelope $\omega$ as in Eq. (4). In App. B we show that our prediction of local frames is indeed equivariant. In App. D, we experimentally investigate the robustness of the local frame estimation to input noise and compare learned local frames against PCA-based local frames.

**Invariance of node features expressed in local frames.** The $3 \times 3$ orthogonal matrix which transforms a vector from the global reference frame into the local frame at node $i$ is given by

$$R_i = (\mathbf{n}_{i,1}, \mathbf{n}_{i,2}, \mathbf{n}_{i,3})^{\mathrm{T}} = \begin{pmatrix} -\mathbf{n}_{i,1}- \\ -\mathbf{n}_{i,2}- \\ -\mathbf{n}_{i,3}- \end{pmatrix}.$$

(6)

This can be easily seen from $R_i \mathbf{n}_{i,1} = (1,0,0)^{\mathrm{T}}$ etc., due to the orthonormality. Under any global transformation $\hat{R} \in \mathrm{O}(3)$ the node positions $\mathbf{x}_i$ and local frame basis vectors $\mathbf{n}_{i,k}$ transform like vectors. The geometric input node features, denoted by $F_i$, transform according to their representation denoted by $\rho_{\text{in}}$. The representation of the input features is determined by the problem setup and may be a combination of scalars, vectors and tensorial features. In formulae, we have

$$\mathbf{x}_i' = \hat{R} \mathbf{x}_i, \quad \mathbf{n}_{i,k}' = \hat{R} \mathbf{n}_{i,k} \quad \forall i, \ k = 1, 2, 3 \quad \text{and} \quad F_i' = \rho_{\text{in}}(\hat{R}) F_i,$$

(7)

which implies the following transformation rule for the local frame $R_i$:

$$R_i' = R_i \hat{R}^{\mathrm{T}} = R_i \hat{R}^{-1} \text{ , since}$$

(8)

$$[R_i']_{mn} = [(\mathbf{n}_{i,1}', \mathbf{n}_{i,2}', \mathbf{n}_{i,3}')^{\mathrm{T}}]_{mn} = (\mathbf{n}_{i,m}')_n = [\hat{R}]_{nl}(\mathbf{n}_{i,m})_l = (\mathbf{n}_{i,m})_l [\hat{R}^{\mathrm{T}}]_{ln} = [R_i \hat{R}^{\mathrm{T}}]_{mn}.$$

In components $(\mathbf{n}_{i,m})_l$ denotes the $l$-th component of the $m$-th basis vector at node $i$. Using Eq. (7) and (8), we can now show that node features expressed in local frames are indeed invariant w.r.t. transformations of the inputs. The coordinates of the global features $F_i$, expressed in the corresponding local frame $R_i$, are given by $\rho_{\text{in}}(R_i)F_i =: f_i$ and the invariance follows as

$$f_i' = \rho_{\text{in}}(R_i')F_i' = \rho_{\text{in}}(R_i \hat{R}^{-1})F_i' = \rho_{\text{in}}(R_i)\rho_{\text{in}}(\hat{R}^{-1})\rho_{\text{in}}(\hat{R})F_i = \rho_{\text{in}}(R_i)F_i = f_i.$$

(9)

**Equivariance of the output.** After transforming the node features into the local frames, the node features can be processed and combined using arbitrary functions during message passing, such as standard MLPs, non-linearities and norm layers. Afterwards, the invariant features $f_i$ are transformed back into the global frame to obtain an equivariant prediction $\rho_{\text{out}}(R_i^{-1})f_i =: Y_i$. The output representation is determined by the problem setup, e.g. when predicting invariant quantities the output representation is the trivial representation $\rho_{\text{out}}(R) = id$, for vectorial output $\rho_{\text{out}}(R) = R$, etc. Indeed, the final prediction transforms equivariantly under any global transformation $\hat{R}$:

$$Y_i' = \rho_{\text{out}}(R_i'^{-1})f_i' = \rho_{\text{out}}((R_i \hat{R}^{-1})^{-1})f_i = \rho_{\text{out}}(\hat{R})\rho_{\text{out}}(R_i^{-1})f_i = \rho_{\text{out}}(\hat{R})Y_i.$$

(10)

Crucially, Eq. (10) holds for any representation of the output $\rho_{\text{out}}$. This means that, after applying multiple message passing layers to the invariant node features, it is merely a matter of interpretation to decide which numbers shall be combined into a common geometric object and which object it should be. Therefore, our pipeline allows for an equivariant prediction of any geometric object and the output representation can be chosen as required by the problem at hand.

## 4.2 TENSORIAL MESSAGE PASSING

So far, we have seen how to achieve equivariance by transforming into equivariant local frames and then performing message passing on the invariant node features $f_i$. In a general form, the invariant message passing in layer $k$ *without tensorial messages* may be written as

$$f_i^{(k)} = \psi^{(k)}\left(f_i^{(k-1)}, \bigoplus_{j \in \mathcal{N}(i)} \phi^{(k)}\left(f_i^{(k-1)}, f_j^{(k-1)}, \rho_{\text{e}}(R_i)e_{ij}, R_i(\mathbf{x}_i - \mathbf{x}_j)\right)\right),$$

(11)

where $\psi^{(k)}$ and $\phi^{(k)}$ are arbitrary non-linear functions and $\bigoplus_{j \in \mathcal{N}(i)}$ denotes the aggregation over neighbors of node $i$. The message passing defined by Eq. (11) differs from vanilla message passing only by the transformation of the edge vectors $\mathbf{x}_i - \mathbf{x}_j$ and the input edge features $e_{ij}$, whose representation we denote by $\rho_e$. Both expressions $\rho_e(R_i)e_{ji}$ and $R_i(\mathbf{x}_i - \mathbf{x}_j)$ are invariant under global transformations as instances of Eq. (9). Together with the invariance of the node features, this guarantees the invariance of Eq. (11). Furthermore, translation invariance is realized trivially by operating only on relative node positions.

The message passing of previous works like (Luo et al., 2022) can be expressed in the form of Eq. (11). However, it has important implications that the invariant node features are expressed in *different* local frames: message passing in the form of Eq. (11) does not allow for the direct communication of geometric information. In some cases, the network may need to communicate directional information, which transforms equivariantly (e.g. encoded in a vector). However, using Eq. (11), the message $f_j$ received by node $i$ is expressed in coordinates of the sending node $j$. Now, since $f_j$ describes an equivariant geometric feature in the neighborhood of node $j$ and since the coordinate frame $R_j$ also transforms equivariantly, the local coordinates $f_j$ are invariant under local transforms (as in Eq. (9) for global transforms). Consequently, node $i$ receives the same message, irrespective of local transformations of the geometry around node $j$ and the global direction of the feature $f_j$ is not preserved (as illustrated in Fig. 1). Our framework remedies exactly this weakness by incorporating proper tensorial messages in the message passing between local frames. Indeed, by including the change of basis in the message passing (cf. $\rho_f(R_i R_j^{-1})f_j$ in Eq. (12)), the messages do preserve the global direction of the feature.

In the final transformation from the local frames back to the global reference frame, the invariant features $f_i$ are interpreted as coordinates of geometric objects. The key idea of our framework is to include exactly such transformations already during message passing. As part of the architecture, one chooses the transformation behavior of $f_i$ in every layer as a direct sum of multiple tensor and pseudotensor representations (see App. A). This combined representation, denoted by $\rho_f$, defines how $f_i$ is transformed from one local frame to the other, based on Eq. (2) and (3). During training, the network learns to communicate vectorial and tensorial information through the respective feature channels, simply because they transform accordingly. That is, if a node would like to send a direction in the form of a vector to its neighbor, it will store the three respective coordinates in three channels of $f_i$ which by design transform like a vector (as illustrated in Fig. 3). As our main result, this yields the following strictly more general form of invariant message passing *with tensorial messages* between local frames:

$$f_i^{(k)} = \psi^{(k)}\left( f_i^{(k-1)}, \bigoplus_{j \in \mathcal{N}(i)} \phi^{(k)}\left( f_i^{(k-1)}, \rho_f(R_i R_j^{-1})f_j^{(k-1)}, \rho_e(R_i)e_{ji}, R_i(\mathbf{x}_i - \mathbf{x}_j) \right) \right). \quad (12)$$

Indeed, the transformation of an invariant node feature $f_j$ from the local frame of node $j$ into the one of node $i$ is also invariant:

$$\rho_f(R_i R_j^{-1})f_j \xrightarrow{\text{global } \hat{R}} \rho_f((R_i \hat{R}^{-1})(R_j \hat{R}^{-1})^{-1})f_j' = \rho_f(R_i \hat{R}^{-1} \hat{R} R_j^{-1})f_j = \rho_f(R_i R_j^{-1})f_j. \quad (13)$$

The formalism in Eq. (12) allows modifying all existing message passing approaches of this form to be $\mathrm{O}(d)$-equivariant and communicate tensorial messages of arbitrary representations. This highlights once more that $\rho_f$ can be chosen freely at every message passing layer as part of the architecture. In practice, we opt for the (pseudo-)tensor representations as feature representations in our networks since they can be implemented efficiently directly using the transformation matrices of the local frames, e.g. by utilizing highly optimized Einstein summation algorithms (cf. Eq. (2) and (3)).

### 4.3 RELATION TO DATA AUGMENTATION

As a direct consequence of condition (1), any group representation maps the identity element to the identity matrix. Thus, if all local frames were chosen to be the identity, Eq. (12) would simply turn into the usual non-invariant message passing. Similarly, choosing all local frames to be the same group element, i.e. $R_i = \tilde{R}$, $\tilde{R} \in \mathrm{O}(d) \, \forall i$, is equivalent to a global transformation of the input data. In this case, tensorial messages do not transform when sent from one local frame to another, since

the change of basis $R_i R_j^{-1}$ is trivial. Therefore, choosing $\tilde{R} \in \mathrm{O}(d)$ randomly for every training sample precisely amounts to data augmentation with random global rotations and reflections.

One clear advantage of our framework is that it allows for a direct comparison between equivariant message passing and data augmentation. Essentially all other works that present an equivariant pipeline do not compare against a non-equivariant baseline trained with data augmentation. Presumably, this is due to the fact that these approaches use equivariant building blocks, which do not have a straightforward non-equivariant equivalent. In our case, the architecture can be trained in both ways for a fair comparison using the same hyperparameters in architecture and optimizer.

### 4.3.1 REFINING THE LOCAL FRAMES DURING MESSAGE PASSING

Meaningful local frames do facilitate expressive local canonicalization, as we demonstrate experimentally (see Tab. 3). Conceptually, this has the following reason: The choice of local frames influences the coordinates in which geometric features are expressed and thereby the computations performed in the non-linear functions (typically MLPs) $\phi^{(k)}$ (cf. Eq. (12)). These embed the incoming geometric information. When choosing the local frames in a systematic manner, adjacent nodes tend to have similar local frames so that similar geometric features are represented using similar coordinates. This facilitates the exchange and processing of information.

The prediction of the local frames, defined by Eq. (4), considers the initial node features and the local geometry in a fairly simple way and only up to a finite cutoff radius. We therefore propose to refine the local frames during the message passing procedure as the field of view grows and more geometric information is aggregated. Using the invariant node features $f_i^{(k)}$ at node $i$ and layer $k$, we employ a simple MLP to predict 6 numbers. These are interpreted as components of two vectors $\mathbf{v}_{i,1}^{(k)}$ and $\mathbf{v}_{i,2}^{(k)}$. Which are then used to generate an SO(3)-matrix $U_i^{(k)}$ by the Gram-Schmidt procedure similar to Sec. 4.1. For every node $i$, the current local frame is then updated according to

$$R_i^{(k)} = U_i^{(k)} R_i^{(k-1)}. \tag{14}$$

The handedness of $R_i$ is preserved, since $U \in \mathrm{SO}(3)$. Note that the $R_i$ are global (equivariant) objects while $U$ is a local (invariant) object so that $R_i^{(k)}$ has the same transformation behavior as $R_i^{(k-1)}$. To ensure that the node features $f_i^{(k)}$ still represent the same geometric objects, they must be transformed from the old local frame into the new one:

$$f_i^{(k)} \to \rho_{\mathrm{f}} \left( (U_i^{(k)} R_i^{(k-1)}) (R_i^{(k-1)})^{-1} \right) f_i^{(k)} = \rho_{\mathrm{f}} \left( U_i^{(k)} \right) f_i^{(k)}. \tag{15}$$

## 5 EXPERIMENTS

Below, we present results for two point cloud experiments using the popular, non-equivariant Point-Net++ architecture (Qi et al., 2017b), adapted to our equivariant framework. Similar to a U-Net for images (Ronneberger et al., 2015), the PointNet++ architecture combines an encoder that iteratively subsamples the point cloud with a decoder that afterwards upsamples the point cloud again. The message passing in the encoder and decoder both follow the form of Eq. (12). Compared to the original PointNet++, we make minor architectural changes, e.g. by introducing radial and angular embeddings (see App. C and E for details of the architecture and training setup).

We have trained different variants of our PointNet++ adaptation on normal vector regression and classification on the ModelNet40 dataset (Wu et al., 2015) and on segmentation on the ShapeNet dataset (Yi et al., 2016). ModelNet40 consists of 12,311 3D shapes of 40 different categories. We use the resampled version of the dataset for which normal vectors at all points are available and use the default train/test split. The ShapeNet dataset consists of around 17,000 3D point clouds (including normal vectors) from 16 shape categories, annotated with 50 semantic classes for segmentation. For all tasks, we compare an equivariant model that uses the tensorial message passing approach (Eq. (12)) against an equivariant model which uses the less general scalar message passing (Eq (11)). In both models, the local frames are learned via Eq. (4) and we optionally include iterative refinement of the local frames (cf. Sec. 4.3.1). Further, we compare against a PointNet++ variant trained with data augmentation (as described in Sec. 4.3). For all three models, we use the same hyperparameters in architecture and optimizer. The main results can be found in Tab. 1

Table 1: **Normal vector regression on ModelNet40.** We report cosine similarities (higher is better) between predicted and target normal vectors for three scenarios: $(z/z)$ trained and evaluated with augmentations around the gravitational axis, $(z/\mathrm{SO}(3))$ trained only with rotations around $z$ but evaluated using all transforms of $\mathrm{O}(3)$ or $\mathrm{SO}(3)$ and $(\mathrm{SO}(3)/\mathrm{SO}(3))$ trained and evaluated using all transforms. Our equivariant adaptation of PointNet++ (Qi et al., 2017b) produces state-of-the-art results. The iterative refinement of the local frames (Sec. 4.3.1) further improves the model. Results of related works are taken from (Luo et al., 2022).

| Method | $z/z$ | $z/\mathrm{SO}(3)$ | $\mathrm{SO}(3)/\mathrm{SO}(3)$ | equivariant |
|---|---|---|---|---|
| RS-CNN (Liu et al., 2019) | 0.74 | 0.17 | 0.50 | ✗ |
| DGCNN (Wang et al., 2019) | 0.71 | 0.68 | 0.78 | ✗ |
| GC-Conv (Zhang et al., 2020) | 0.58 | 0.56 | 0.58 | ✓ |
| Luo et al. (Luo et al., 2022) | 0.80 | 0.80 | 0.80 | ✓ |
| Method | $z/z$ | $z/\mathrm{O}(3)$ | $\mathrm{O}(3)/\mathrm{O}(3)$ | equivariant |
| Data augmentation | **0.89** | 0.79 | 0.86 | ✗ |
| Learned frames + scalar messages (ours) | 0.82 | 0.82 | 0.82 | ✓ |
| Learned frames + refining frames + scalar messages (ours) | 0.83 | 0.83 | 0.83 | ✓ |
| Learned frames + tensor messages (ours) | 0.87 | 0.87 | 0.87 | ✓ |
| Learned frames + refining frames + tensor messages (ours) | 0.88 | **0.88** | **0.88** | ✓ |

Table 2: **Segmentation on ShapeNet.** Our equivariant adaptation of PointNet++ (Qi et al., 2017b) yields competitive results and significantly outperforms the corresponding model trained with data augmentation. Both tensorial messages and iterative refinement of the local frames enhance the performance. Training and evaluation setups as in Tab. 1. Results of related works are based on the original papers and on (Luo et al., 2022; Lou et al., 2023; Deng et al., 2021).

| Method | $z/z$ | $z/\mathrm{SO}(3)$ | $\mathrm{SO}(3)/\mathrm{SO}(3)$ | invariant |
|---|---|---|---|---|
| PointNet (Qi et al., 2017a) | 76.2 | 37.8 | 74.4 | ✗ |
| DGCNN (Wang et al., 2019) | 78.8 | 37.4 | 73.3 | ✗ |
| RI-Conv (Zhang et al., 2019) | 75.6 | 75.3 | 75.5 | ✓ |
| LGR-Net (Zhao et al., 2022) | 80.0 | 80.0 | 80.1 | ✓ |
| Li et al. (w/ TTA) (Li et al., 2021a) | 75.9 | 75.9 | 75.9 | ✓ |
| CRIN (Lou et al., 2023) | 80.5 | 80.5 | 80.5 | ✓ |
| Luo et al. (Luo et al., 2022) | - | 80.9 | 80.8 | ✓ |
| TFN (Thomas et al., 2018) | - | 76.8 | 76.2 | ✓ |
| VN-PointNet (Deng et al., 2021) | - | 72.4 | 72.8 | ✓ |
| VN-DGCNN (Deng et al., 2021) | - | **81.4** | **81.4** | ✓ |
| Method | $z/z$ | $z/\mathrm{O}(3)$ | $\mathrm{O}(3)/\mathrm{O}(3)$ | invariant |
| Data augmentation | **81.9** | 12.5 | 78.0 | ✗ |
| Learned frames + scalar messages (ours) | 78.8 | 78.8 | 78.8 | ✓ |
| Learned frames + refining frames + scalar messages (ours) | 79.0 | 79.0 | 79.0 | ✓ |
| Learned frames + tensor messages (ours) | 79.7 | 79.7 | 79.7 | ✓ |
| Learned frames + refining frames + tensor messages (ours) | 80.2 | 80.2 | 80.2 | ✓ |

and 2. For additional experimental results, including experiments on the real-world dataset ScanObjectNN (Uy et al., 2019), see App. D. The model trained with data augmentation has slightly fewer learnable parameters since the local frames are not learned but chosen randomly for data augmentation (cf. Sec. 4.3). For the normal regression model (with iteratively refined local frames), this difference amounts to 0.33% (9.1%), for the segmentation model to 0.11% (10.3%) and for classification to 0.14% (3.9%).

Table 3: **Learned informative local frames and tensorial messages are beneficial.** Normal vector regression on ModelNet40 using adaptations of PointNet++ with learned vs. random local frames and tensorial vs. scalar messages (without local frame refinement). Tensorial messages significantly improve the performance. Even with randomly chosen local frames, the model with tensorial messages outperforms both models with scalar messages.

| Cosine similarity ↑ | tensorial messages | scalar messages |
|---|---|---|
| learned local frames | **0.87** | 0.82 |
| random local frames | 0.84 | 0.80 |

## 6    DISCUSSION

On normal vector regression, our equivariant adaptation of PointNet++ with tensorial messages achieves state-of-the-art results.    Normal regression is a task in which equivariance is certainly desirable and in which geometric information must be propagated precisely.   Thus, tensorial message passing proves to be superior over scalar messages due to the limitations mentioned in Sec. 4.2 and Fig. 1. For point cloud segmentation the gain is less significant, indicating that geometric information, e.g. in the form of characteristic directions, may be less important.   In both experiments, the networks that exhibit exact equivariance yield substantial improvements over data augmentation.   Moreover, we demonstrate that informative local frames are indeed beneficial through an ablation study with randomly chosen local frames (see Tab. 3). We find that the model with tensor messages significantly

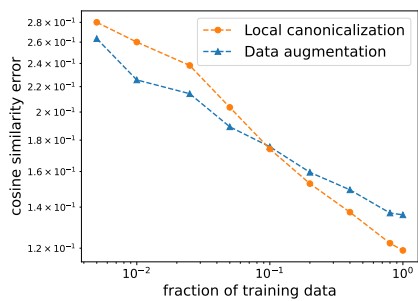

Figure 4: **Data efficiency of built-in equivariance vs. data augmentation.**

outperforms the model with scalar messages (even if the local frames are chosen randomly). This highlights once more that the tensorial message passing approach enables to communicate geometric information more reliably.

Since equivariant methods do neither "waste" data nor network capacity to perform well on different input orientations, they are often said to be more data efficient (Batzner et al., 2022), meaning that they improve faster as more data becomes available (Hestness et al., 2017). We have trained a series of networks on normal regression (as in Sec. 5) with the same architecture but different fractions of the training data. According to (Hestness et al., 2017), the error rate as function of the training set size typically takes the form of a power-law. Indeed, in the log-log plot (test cosine similarity error vs. fraction of training data, Fig. 4) the equivariant approach shows a steeper slope, indicating better data efficiency than the same model trained with data augmentation. However, perhaps surprisingly, the error rate is not necessarily smaller for all dataset sizes, meaning that in some cases, purely in terms of accuracy, data augmentation may be favorable over built-in equivariance. This result connects to recent research studying the effects of learning approximate symmetries (Langer et al., 2024) and the question whether equivariance verifiably improves the scaling of neural networks (Brehmer et al., 2024). We see this as an advantage of our framework, which allows parallel development of an exact equivariant model and an equally well-engineered non-equivariant baseline trained via data augmentation.

## 7    CONCLUSION

This work introduces a novel message passing formalism, which together with local canonicalization forms a framework for building $O(d)$-equivariant message passing architectures. We provide a well-motivated formalism through which existing non-equivariant networks can be adapted to be equivariant. The presented approach can be integrated straightforwardly with any existing message passing architecture. One the one hand, our approach provides a new perspective contrary to most existing approaches for exact equivariance, which do not use local frames but specialized tensorial operations. On the other hand, our method offers a strict generalization of approaches that do achieve equivariance based on local canonicalization. We demonstrate that local canonicalization paired with tensorial messages can significantly improve the performance compared to methods that use local canonicalization without tensorial messages. Our framework can be used as a drop-in replacement for data augmentation to achieve exact, built-in equivariance and allows for a direct and fair comparison between the two approaches. As such, our approach opens up a new possibility for evaluating the efficacy of equivariance as a model prior on numerous geometric machine learning tasks. Based on the broad applicability of our formalism, we hope to inspire researchers and practitioners alike to utilize this approach in many different architectures in various domains.

ACKNOWLEDGMENTS

This work is supported by the Klaus Tschira Stiftung gGmbH (SIMPLAIX project P4) and by Deutsche Forschungsgemeinschaft (DFG) under Germany's Excellence Strategy EXC-2181/1 - 390900948 (the Heidelberg STRUCTURES Excellence Cluster) as well as under project number 240245660 - SFB 1129.

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

## A  TENSOR REPRESENTATION

The tensor representation is introduced in Eq. 2. The tensor representation is a group representation of $O(d)$ and is used in this paper to define the transformation behavior of the tensorial objects in the tensorial message passing.

**Proof representation property.**  The representation property follows from the fact that the $3 \times 3$ matrices are a representation. For any $R_1, R_2 \in O(d)$ we have

$$
\begin{aligned}
[\rho(R_1)\rho(R_2)T]_{i_1 \dots i_n} &= R_{1,i_1 k_1} \dots R_{1,i_n k_n} R_{2,k_1 j_1} \dots R_{2,k_n j_n} T_{j_1 \dots j_n} \\
&= [R_1 R_2]_{i_1 j_1} \dots [R_1 R_2]_{i_n j_n} T_{j_1 \dots j_n} = [\rho(R_1 R_2)T]_{i_1 \dots i_n}
\end{aligned}
\tag{16}
$$

The pseudo-vector representation is also a representation, which can be shown similarly by using that the determinant is multiplicative: $\det(AB) = \det A \det B$.

**Representation hidden features.**  The input and output representations are determined by the input data and the prediction task. Contrarily, the internal representations are chosen as hyperparameters. In our framework, we choose a direct sum of tensor representations, which again forms a representation. A feature $f$ will transform under $\rho_f = \rho_1 \oplus \dots \oplus \rho_k$ with a block diagonal matrix:

$$
\rho_f(R)f = \begin{pmatrix} \rho_1(R) & & \\ & \ddots & \\ & & \rho_k(R) \end{pmatrix} f
\tag{17}
$$

So that the composition of scalar, vectorial and tensorial features can be chosen freely. The feature dimension of $f$ will be given by $\dim(f) = \dim(\rho_1) + \dots + \dim(\rho_k)$.

## B  LEARNING LOCAL FRAMES

Learning the local frames is an essential part of the proposed architecture, therefore we present some further considerations for predicting local frames in an $O(3)$ equivariant way.

As described in Sec. 4.1, for each node $i$ one predicts two vectors $\mathbf{v}_{i,1}$ and $\mathbf{v}_{i,2}$ by summing over the relative positions of adjacent nodes, weighted by the output of an MLP and an envelope function $\omega$. The envelope function, adapted from (Gasteiger et al., 2020), is given by

$$
w(r_{ij}) = \begin{cases} 1 - \frac{(p+1)(p+2)}{2}\left(\frac{r_{ij}}{r_c}\right)^p + p(p+2)\left(\frac{r_{ij}}{r_c}\right)^{p+1} - \frac{p(p+1)}{2}\left(\frac{r_{ij}}{r_c}\right)^{p+2} & r_{ij} < r_c \\ 0 & r_{ij} \geq r_c \end{cases}
\tag{18}
$$

and ensures a smooth transition at the cutoff radius $r_c$ of the local neighborhood. Here, $r_{ij} = \|\mathbf{x}_j - \mathbf{x}_i\|$ is the relative distance between the nodes. The parameter $p$ is chosen to be 5 for all experiments presented in this paper. Afterwards, the two vectors $\mathbf{v}_{i,1}, \mathbf{v}_{i,2}$ are used to construct two orthonormal vectors $\mathbf{n}_{i,1}, \mathbf{n}_{i,2}$ using the Gram-Schmidt procedure:

$$
\mathbf{n}_{i,1} = \frac{\mathbf{v}_{i,1}}{\|\mathbf{v}_{i,1}\|}
\tag{19}
$$

$$
\mathbf{n}'_{i,2} = \mathbf{v}_{i,2} - (\mathbf{n}_{i,1} \cdot \mathbf{v}_{i,2})\, \mathbf{n}_{i,1}
\tag{20}
$$

$$
\mathbf{n}_{i,2} = \frac{\mathbf{n}'_{i,2}}{\|\mathbf{n}'_{i,2}\|}
\tag{21}
$$

The third vector is chosen to point in the same half-space as the local center of mass to ensure an $O(3)$-equivariant construction. The estimate of the direction to the local center of mass is smoothed using the same envelope function $\omega$.

$$
\mathbf{n}_{i,3} = \begin{cases} \mathbf{n}_{i,1} \times \mathbf{n}_{i,2} & \text{if} \quad (\mathbf{n}_{i,1} \times \mathbf{n}_{i,2}) \cdot \bar{\mathbf{r}} \geq 0 \\ -\mathbf{n}_{i,1} \times \mathbf{n}_{i,2} & \text{else} \end{cases} \quad , \text{ with } \bar{\mathbf{r}} := \sum_{j \in \mathcal{N}(i)} \omega(r_{ij})(\mathbf{x}_j - \mathbf{x}_i),
\tag{22}
$$

The third vector may not be learned. If one constrains the local coordinate frames to be orthonormal, the third vector is defined up to its sign; and predicting this sign is a non-differentiable operation.

Our experiments have shown that orthonormal frames, which are associated with O(3) transformations, are favorable and that a relaxation of the normalization or orthogonality of the basis vectors decreases the performance of the models.

It is worth mentioning that despite the smooth envelope function canonical frames can never be fully continuous, e.g. in highly symmetric cases. In the unlikely case, that $\mathbf{v}_1$ and $\mathbf{v}_2$ are parallel we sample the direction of $\mathbf{v}_2$ randomly.

Lastly, let us briefly show that our prediction of local frames is indeed equivariant, meaning that they transform consistently as the input point cloud is flipped or rotated. That is, we need to show that $\mathbf{n}_{i,k} \to \mathbf{n}'_{i,k} = \hat{R}\mathbf{n}_{i,k}$ under any global transformation $\hat{R} \in$ O(3). Clearly, $\mathbf{v}_{i,1}$ and $\mathbf{v}_{i,2}$ transform like vectors, i.e. $\mathbf{v}'_{i,k} = \hat{R}\mathbf{v}_{i,k}$ for $k = 1, 2$, since they are constructed as a weighted sum of node positions (cf. Eq. (4)). The same holds true for $\bar{\mathbf{r}}$ and for the two basis vectors $\mathbf{n}_{i,1}$ and $\mathbf{n}_{i,2}$. As a consequence, the third basis vector $\mathbf{n}_{i,3}$ transforms as

$$\mathbf{n}_{i,3} \to \mathbf{n}'_{i,3} = \begin{cases} (R\mathbf{n}_{i,1}) \times (R\mathbf{n}_{i,2}) & \text{if} \quad \big((R\mathbf{n}_{i,1}) \times (R\mathbf{n}_{i,2})\big) \cdot R\bar{\mathbf{r}} > 0 \\ -(R\mathbf{n}_{i,1}) \times (R\mathbf{n}_{i,2}) & \text{else} \end{cases} \tag{23}$$

$$= R \begin{cases} \det(R)(\mathbf{n}_{i,1} \times \mathbf{n}_{i,2}) & \text{if} \quad \det(R)(\mathbf{n}_{i,1} \times \mathbf{n}_{i,2}) \cdot \bar{\mathbf{r}} > 0 \\ -\det(R)(\mathbf{n}_{i,1} \times \mathbf{n}_{i,2}) & \text{else} \end{cases} \tag{24}$$

$$= R\mathbf{n}_{i,3}, \tag{25}$$

since $(R\mathbf{v}) \times (R\mathbf{w}) = \det(R)R(\mathbf{v} \times \mathbf{w})$.

## C  EQUIVARIANT POINTNET++ USING OUR FRAMEWORK

PointNet++ is a widely-used architecture for point cloud tasks. It combines an encoder that iteratively down samples the point cloud with a decoder with upsampling (Qi et al., 2017b). Each layer in the encoder consists of the following steps:

1. Use furthest point sampling to sample a subset of equally spaced nodes $N^{(k)}$.
2. For each node $i$ in $N^{(k)}$ generate its neighborhood $\mathcal{N}(i)$ by finding all nodes within a specified radius $r_{max}^{(k)}$.
3. Send and aggregate messages from all neighbors according to:

$$f_i^{(k,enc)} = \max_{j \in \mathcal{N}(i)} \phi\left(f_j^{(k-1,enc)}, \mathbf{x}_j - \mathbf{x}_i\right), \tag{26}$$

   with the channel-wise maximum as an aggregation function.
4. Continue with the next layer, but keep only the nodes $N^{(k)}$.

Since Eq. (26) follows precisely the form of Eq. (12), the message passing formula can easily be adapted to our formalism by

$$f_i^{(k,enc)} = \max_{j \in \mathcal{N}(i)} \phi\left(\rho_{\mathrm{f}}(R_i R_j^{-1}) f_j^{(k-1,enc)}, R_i(\mathbf{x}_j - \mathbf{x}_i)\right), \tag{27}$$

where $R_i$ is the local frame of node $i$ and $\rho_{\mathrm{f}}(g_i g_j^{-1})$ the representation under which the node features are transformed from the local frame of node $j$ into the one at node $i$. We further refine the messages by splitting the edge vectors $R_i(\mathbf{x}_j - \mathbf{x}_i)$ into a radial and an angular embedding. For the angular embedding, we simply use the normalized direction. For the norm of the edge vector, we use a Gaussian embedding, with $k$ Gaussians-like functions, with means $\mu_k$ equidistantly spaced in the interval $[0, r_{max}]$ and standard deviation $\sigma_k$ so that adjacent Gaussians intersect at a function value of 0.5:

$$(\tilde{r}_{ij})_k = \exp\left(-\frac{(\|\mathbf{x}_j - \mathbf{x}_i\| - \mu_k)^2}{2\sigma_k^2}\right) \tag{28}$$

The complete, invariant message passing step reads

$$f_i^{(k,enc)} = \max_{j \in \mathcal{N}(i)} \phi\left(\rho_{\mathrm{f}}(R_i R_j^{-1}) f_j^{(k-1,enc)}, \tilde{r}_{ij}, R_i \frac{\mathbf{x}_j - \mathbf{x}_i}{\|\mathbf{x}_j - \mathbf{x}_i\|}\right). \tag{29}$$

If the task requires one output for the entire point cloud, the node features at the nodes, remaining after the encoder $N^{(k^*)}$, are pooled into one global feature. The node and the local frame to which one sends all these messages is the one that is closest to the center of mass of the point cloud, i.e.:

$$\hat{\imath} = \arg\max_{i \in N^{(k^*)}} \|\mathbf{x}_i - \bar{\mathbf{x}}\| \text{ with } \bar{\mathbf{x}} := \sum_{i \in N^{(k^*)}} \mathbf{x}_i \bigg/ \sum_{i \in N^{(k^*)}} 1 \qquad (30)$$

$$f_{global} = \max_{j \in \mathcal{N}_{\hat{\imath}}} \phi \left( \rho_{\mathrm{f}}(R_{\hat{\imath}} R_j^{-1}) f_j^{(k_{max},enc)}, \tilde{r}_{\hat{\imath}j}, R_{\hat{\imath}} \frac{\mathbf{x}_j - \mathbf{x}_{\hat{\imath}}}{\|\mathbf{x}_j - \mathbf{x}_{\hat{\imath}}\|} \right) \qquad (31)$$

Finally, the global features may be passed through an MLP to generate the output of the invariant message passing part of our architecture.

If the task requires one output per node in the input point cloud, one must upsample the nodes again after the encoder. To do this, one caches the positions and features of the nodes of the encoder layers and iteratively applies the following steps in the decoder:

1. Let the input nodes to that layer be $N^{(k)}$. The features at these nodes are interpolated to the node features of the larger subset $N^{(k-1)}$ (reversing the subsampling of the encoder layers): For that, one finds for each node $i \in N^{(k-1)}$ its three closest neighbors in $N^{(k)}$ (let us denote their set by $\mathrm{NN}_3(i)$) and interpolates their features by inverse distance weighting:

$$h_i = \frac{\sum_{j \in \mathrm{NN}_3(i)} \|\mathbf{x}_j - \mathbf{x}_i\|^{-1} \rho_{\mathrm{f}}(R_i R_j^{-1}) f_j^{(k,dec)}}{\sum_{j \in \mathrm{NN}_3(i)} \|\mathbf{x}_j - \mathbf{x}_i\|^{-1}}. \qquad (32)$$

2. The interpolated features $h_i$ are then concatenated with the node features at the $k-1$-th layer of the encoder and embedded in an MLP to obtain the updated and upsampled node features:

$$f_i^{(k-1,dec)} = \mathrm{MLP}\left(h_i, f_i^{(k-1,enc)}\right), \qquad (33)$$

which practically implements a skip connection between the activations in the encoder and the decoder. Note that the decoder features are counted backward in $k$.

3. Continue with the next layer with the set of nodes given by $N^{(k-1)}$.

Finally, the node features, back at the level of the input nodes, are brought into the desired output dimension by one last MLP layer, before they are transformed into the global frame of reference to obtain an equivariant prediction.

# D   ADDITIONAL EXPERIMENTS

**Classification on ModelNet40.**   As another experiment, we trained a set of models on the classification task on ModelNet40 (with variants as for normal regression). Classification requires a global output at the point cloud level so we use only the encoder of PointNet++ architecture (as described in App. C). The results of this experiment can be found in Tab. 4. Considering full $O(3)$-equivariance, we find that the models with tensorial message passing outperform the corresponding model trained with data augmentation and scalar message passing. As for the other experiments, the best-performing model is the one with local frames that are learned and iteratively refined. While our adaptations of the fairly simple PointNet++ architecture do not yield state-of-the-art results in this task, our approach boosts PointNet++ to still be competitive.

**Evaluation runtimes.**   To compare the impact of our proposed message passing formalism on the evaluation time, we measured the interference times of the trained models on the normal vector regression tasks. The results of this ablation can be found in Tab. 5.

**Classification on real-world dataset ScanObjectNN.**   To provide experimental evidence that our framework similarly prevails on a real-world dataset, we conducted a series of experiments on the ScanObjectNN dataset (Uy et al., 2019) (model variants and evaluation setup exactly as in the classification task on ModelNet40). The dataset contains scanned indoor scene data subject to realistic

Table 4: **Classification accuracies on ModelNet40.** Our equivariant adaptation of PointNet++ (Qi et al., 2017b) produces superior results over the vanilla PointNet++ (with training and evaluation setups as in Tab. 1). Results of related works are based on the original papers and on (Luo et al., 2022; Lou et al., 2023; Deng et al., 2021).

| Method | $z/z$ | $z/SO(3)$ | $SO(3)/SO(3)$ | invariant |
|---|---|---|---|---|
| PointNet (Qi et al., 2017a) | 85.9 | 19.6 | 74.7 | ✗ |
| RS-CNN (Liu et al., 2019) | 90.3 | 48.7 | 82.6 | ✗ |
| DGCNN (Wang et al., 2019) | 90.3 | 33.8 | 88.6 | ✗ |
| RI-Conv (Zhang et al., 2019) | 86.5 | 86.4 | 86.4 | ✓ |
| GC-Conv (Zhang et al., 2020) | 89.0 | 89.1 | 89.2 | ✓ |
| Luo et al. DGCNN (Luo et al., 2022) | 88.4 | 88.4 | 88.9 | ✓ |
| LGR-Net (Zhao et al., 2022) | 90.9 | 90.9 | 91.1 | ✓ |
| Li et al. (w/ TTA) (Li et al., 2021a) | 91.6 | 91.6 | 91.6 | ✓ |
| CRIN (Lou et al., 2023) | **91.8** | **91.8** | **91.8** | ✓ |
| TFN (Thomas et al., 2018) | 88.5 | 85.3 | 87.6 | ✓ |
| VN-PointNet (Deng et al., 2021) | 77.5 | 77.5 | 77.2 | ✓ |
| VN-DGCNN (Deng et al., 2021) | 89.5 | 89.5 | 90.2 | ✓ |
| Method | $z/z$ | $z/O(3)$ | $O(3)/O(3)$ | invariant |
| Data augmentation | 89.6 | 16.8 | 86.6 | ✗ |
| Learned frames + scalar messages (ours) | 86.7 | 86.7 | 86.7 | ✓ |
| Learned frames + refining frames + scalar messages (ours) | 87.3 | 87.3 | 87.3 | ✓ |
| Learned frames + tensor messages (ours) | 88.0 | 88.0 | 88.0 | ✓ |
| Learned frames + refining frames + tensor messages (ours) | 88.7 | 88.7 | 88.7 | ✓ |

Table 5: **Evaluation runtimes.** Average runtime for a single sample on normal vector regression (executed on one NVIDIA A100). Standard deviations are based on 10 loops over the test set. The first model achieves the best accuracy and is exactly equivariant, but the runtime of data augmentation is 32% faster.

| Method | evaluation runtime |
|---|---|
| Learned frames + refining frames + tensor messages | $(0.25 \pm 0.07)s$ |
| Learned frames + tensor messages | $(0.22 \pm 0.06)s$ |
| Random frames + tensor messages | $(0.21 \pm 0.06)s$ |
| Learned frames + scalar messages | $(0.20 \pm 0.06)s$ |
| Random frames + scalar messages | $(0.18 \pm 0.06)s$ |
| Data augmentation | $(0.17 \pm 0.06)s$ |

measurement noise and includes deformable objects (e.g. the bag class) and multi-body objects (e.g. a shelf with objects in it). The results of these experiment can be found in Tab. 6. While our adaptation of the PointNet++ architecture is not state-of-the-art, we find that our proposed framework is consistently better in the $O(3)/O(3)$ setup than data augmentation. Furthermore, tensorial messages substantially outperform the networks using scalar message passing.

**Robustness to noise.** To assess the robustness of our approach to noisy inputs, we evaluated both the robustness of the overall architecture and the robustness of the local frame estimation (as described in Sec. 4.1. We evaluated our best-performing models for normal regression and segmentation tasks (learned frames + refining frames + tensor messages) and added Gaussian noise of varying scale $\sigma$ to the node positions (and input point normals for ShapeNet). Furthermore, we have re-trained the two models with input jitter during training (and otherwise identical settings) to study how well these models would learn robustness to noise. For ShapeNet, we find that both models are relatively robust. The segmentation quality (measured by the intersection over union) does not decrease significantly for noise scales up to the average neighbor distance in the dataset (Fig. 5a). Unsurprisingly, the model trained with noisy inputs is slightly more robust. Qualitatively, the models trained on normal regression display the same behavior (Fig. 5b), though in this case the cosine similarity between predicted and ground truth normals drops faster as one increases the noise level (especially for the model trained without noise). We suspect the following reason: the targets in the normal regression dataset are obtained from mesh representations of the CAD models. Without any noise the model may learn to estimate the these normals most accurately by fitting planes very

Table 6: **Classification accuracies on ScanObjectNN.** Our equivariant adaptation of Point-Net++ (Qi et al., 2017b) produces superior results over the vanilla PointNet++ (with training and evaluation setups as in Tab. 1). Results of related works are based on the original papers and on (Lou et al., 2023).

| Method | no augm. | no augm./SO(3) | SO(3)/SO(3) | invariant |
|---|---|---|---|---|
| PointNet (Qi et al., 2017a) | 79.4 | 16.7 | 54.7 | ✗ |
| DGCNN (Wang et al., 2019) | **87.3** | 17.7 | 71.8 | ✗ |
| RI-Conv (Zhang et al., 2019) | - | 78.4 | 78.1 | ✓ |
| LGR-Net (Zhao et al., 2022) | - | 81.2 | 81.4 | ✓ |
| Li et al. (w/ TTA) (Li et al., 2021a) | 86.7 | **86.7** | **86.7** | ✓ |
| CRIN (Lou et al., 2023) | 84.7 | 84.7 | 84.7 | ✓ |
| Method | $z/z$ | $z/O(3)$ | $O(3)/O(3)$ | invariant |
| Data augmentation | 87.0 | 13.3 | 70.3 | ✗ |
| Learned frames + scalar messages (ours) | 71.9 | 71.9 | 71.9 | ✓ |
| Learned frames + refining frames + scalar messages (ours) | 73.7 | 73.7 | 73.7 | ✓ |
| Learned frames + tensor messages (ours) | 79.8 | 79.8 | 79.8 | ✓ |
| Learned frames + refining frames + tensor messages (ours) | 81.0 | 81.0 | 81.0 | ✓ |

locally (to estimate normals from a triangular mesh). Such a very local normal estimation is very much susceptible to noise. The robustness of the local frames is assessed in the following way: for a given input point cloud, estimate the local frames with and without input noise, denoted by $R_i$ and $\tilde{R}_i$ at node $i$ respectively. Then, average the Frobenius norm $\|R_i - \tilde{R}_i\|_F$ over all nodes. Further, we compute the average cosine similarity for the normalized coordinate vectors $R_{i,k} \cdot \tilde{R}_{i,k}$ to investigate the stability of each coordinate axis under noise. For ShapeNet, the local frame estimation is equally robust in both cases. The cosine similarity of the individual coordinate axes is very informative. The first coordinate axis is most robust to noise, indicating that the model learns to (equivariantly) identify a geometrically important direction and predicts it robustly. It is not a coincidence that this axis is the coordinate first axis. The model uses this axis from the most prominent direction, since it will not be changed in the Gram-Schmidt orthogonalization (cf. Sec. 4.1). The other coordinate axes are much more likely to be degenerate due to symmetries (e.g. if points locally form a plane or a perfect sphere). Still, the second axis seems to be often geometrically informative and is robustly estimated in these cases. Figure 5f shows that the robustness of the second component can be improved for normal regression by training with noisy samples. The second and the third component of the local frame are constrained by the orthogonality. Consequently, the deviations from the first (and second) component add up during the Gram-Schmidt orthogonalization, making component two and three less robust.

**PCA-based local frames.** We re-trained our best-performing model on all tasks, replacing the learned local frames with local frames computed using local Principal Component Analysis (PCA) (with the same radial cutoff).

Local cutoff PCA computes a local reference frame for each point $i$ in a point cloud by analyzing the spatial distribution of its neighbors within a defined cutoff radius ($\mathcal{N}(i) = \{j : \|\mathbf{x}_i - \mathbf{x}_j\|_2\}$). For each, node $i$ the local frame is given by the eigenvectors $\mathbf{e}_{i,k}$, $k = 1, 2, 3$ of the local covariance matrix $\sum_{j \in \mathcal{N}(i)} (\mathbf{x}_i - \mathbf{x}_j)(\mathbf{x}_i - \mathbf{x}_j)^\mathrm{T}$. The sign of the eigenvectors is fixed by demanding that $\sum_{j \in \mathcal{N}(i)} \mathbf{e}_{i,k} \cdot (\mathbf{x}_i - \mathbf{x}_j) > 0$. The resulting $\mathbf{e}_{i,k}$, $k = 1, 2, 3$ are O(3)-equivariant and define an orthonormal frame (the extension to O($d$) is trivial).

For the classification task, we observed that PCA-based local frames provided a slight improvement in accuracy compared to the learned frames (88.9 vs. 88.7). Regarding segmentation on ShapeNet, both models achieve the same IOU metric of 80.2. For the normal regression task, the learned and iteratively refined local frames demonstrate superior performance, achieving a higher cosine similarity (0.88 vs. 0.87).

We would like to highlight the flexibility in the choice of local frames as a strength of our formalism. Depending on the task and the structure of the input, different equivariant local frames can be superior and our framework can be readily applied to all equivariant estimators of rule-based or learned local frames.

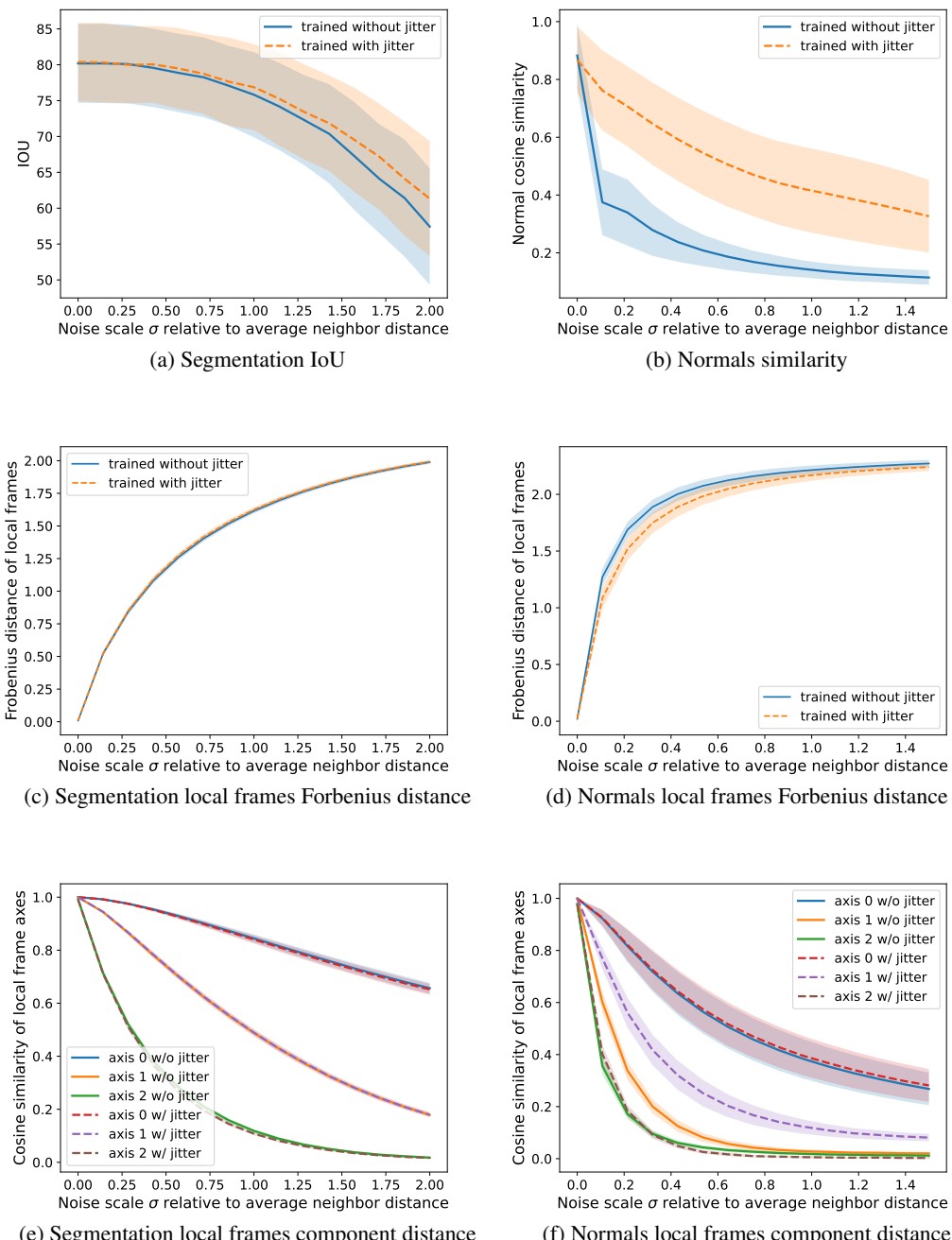

Figure 5: **Robustness analysis on model performance and local frame estimation.** Best-performing models using our framework (learned frames + refining frames + tensor messages) trained with and without input jitter and evaluated on data of varying noise scale. Unsurprisingly, the models trained with jitter are more robust, showing that, in our framework, robustness to noise can be learned from noisy data. The coordinate axes of the local frames are compared against predictions without noise. The first coordinate axis (preserved by Gram-Schmidt) carries most geometric information and is most robust. The second and third coordinate axis may be degenerate in symmetric cases and are consequently less robust. Furthermore, the latter axes also inherit perturbations from the first (and second) axis through the Gram-Schmidt orthogonalization.

# E    EXPERIMENTAL DETAILS

The normal vector regression and the classification experiments are conducted on the ModelNet40 dataset (Wu et al., 2015).    In particular, we use the resampled version available at `https://shapenet.cs.stanford.edu/media/modelnet40_normal_resampled.zip`, which includes normal vectors for each point in the point cloud. We use the first 1024 points based on the ordering provided in this version of the dataset and normalize the point clouds to fit in the unit sphere. The ordering is based on furthest point sampling to evenly cover the surface of the 3D shapes.

The segmentation experiments are conducted on the ShapeNet dataset (Yi et al., 2016). The dataset consists of about 17,000 3D shape point clouds (2048 points and normal vectors each) from 16 shape categories. Each category is annotated with 2 to 6 parts. In total, there are 50 different semantic classes that must be distinguished during the segmentation.

**Hyperparameter choices.**    The hyperparameters chosen for our two main experiments are listed in Tab. 7.

Table 7: **Hyperparameter choices.** The main hyperparameter choices for our models in the classification and normal vector regression task. Label smoothing only applies to the classification and segmentation models. For the classification task on ScanObjectNN we only trained for 500 epochs.

|  | normal vector regression | classification | segmentation |
|---|---|---|---|
| optimizer | AdamW | AdamW | AdamW |
| weight decay | 5e-4 | 0.05 | 1e-3 |
| learning rate | 2.5e-3 | 1e-3 | 0.05 |
| scheduler | Cosine-LR | Cosine-LR | Cosine-LR |
| epochs | 800 | 800 / 500 | 800 |
| warm up epochs | 10 | 10 | 10 |
| gradient clip | 0.5 | 0.5 | 0.5 |
| label smoothing | N.A. | 0.3 | 0.3 |
| loss | L1-loss | Cross-Entropy | Cross-Entropy |

**Architectural design.**    The architectures used in our experiment can be summarized using the following short-hand:

- Encoding layer: $E($`in rep.`$, [\text{hidden layers}], \text{neighborhood radius}, \text{subsampling fraction})$ see Eq. (29)
- Decoding layer: $D($`in rep.`$, [\text{hidden layers}])$ see Eq. (33)
- MLP: `MLP`$($`in rep.`$, [\text{hidden layers}],$ `out rep.`$)$
- Output layer: $O($`in rep.`$, [\text{hidden layers}],$ `out rep.`$, \text{dropout})$ see Eq. (31)

Furthermore, we define the following notation to specify the feature representation used during message passing: The feature representations are a direct sum of tensor and pseudotensor representations. A representation is characterized by its order (i.e. the number of indices, cf. 3) and its behavior under parity (`n` for tensors and `p` for pseudotensors). Furthermore, we specify the multiplicities, that is, how often each representation appears in a direct sum representation. To give an example, the representation denoted as `8x0p+4x1n` is the direct sum of 8 pseudoscalars and 4 vectors.

The architecture used for normal vector regression is described in Tab. 8. The number of Gaussian-like functions in the radial embedding is set to 64. The architecture used for classification is described in Tab. 10. The architecture used for the segmentation of the ShapeNet dataset is described in Tab. 9. For these two experiments, the number of Gaussian-like functions in the radial embedding is set to 16. The MLP used in the prediction of the local frames (according to Eq. (4)) has two hidden layers of dimensions (128, 128) for normal regression and (64, 32) for classification and segmentation. For the iterative refinement of the local frames (Sec. 4.3.1) after each message passing layer we use for all experiments MLPs with hidden dimensions (64, 32).

All fully connected linear layers are followed by batch normalization except the MLP in the output layer, where we do not use any normalization. As activation function, we use the `SiLU` function.

Table 8: **Architecture of the normal vector regression model.**

| Layer number | Layer |
|---|---|
| 1 | $E(\texttt{0x0n}, [64], 0.2, 1.0)$ |
| 2 | $E(\texttt{64x0n+16x0p+16x1n+4x1p+4x2n+1x2p}, [64], 0.2, 1.0)$ |
| 3 | $E(\texttt{64x0n+16x0p+16x1n+4x1p+4x2n+1x2p}, [128], 0.2, 0.2)$ |
| 4 | $E(\texttt{128x0n+32x0p+32x1n+8x1p+8x2n+2x2p}, [256], 0.5, 0.25)$ |
| 5 | $E(\texttt{256x0n+64x0p+64x1n+16x1p+16x2n+4x2p}, [512], 0.8, 0.35)$ |
| 6 | $E(\texttt{512x0n+128x0p+128x1n+32x1p+32x2n+8x2p}, [512], 1.4, 0.5)$ |
| 7 | $D(\texttt{512x0n+128x0p+128x1n+32x1p+32x2n+8x2p}, [512])$ |
| 8 | $D(\texttt{512x0n+128x0p+128x1n+32x1p+32x2n+8x2p}, [256])$ |
| 9 | $D(\texttt{256x0n+64x0p+64x1n+16x1p+16x2n+4x2p}, [128])$ |
| 10 | $D(\texttt{128x0n+32x0p+32x1n+8x1p+8x2n+2x2p}, [128])$ |
| 11 | $D(\texttt{64x0n+16x0p+16x1n+4x1p+4x2n+1x2p}, [64])$ |
| 12 | $D(\texttt{64x0n+16x0p+16x1n+4x1p+4x2n+1x2p}, [64])$ |
| 13 | $\texttt{MLP}(\texttt{64x0n+16x0p+16x1n+4x1p+4x2n+1x2p}, [128, 64, 32], \texttt{1x1n})$ |

Table 9: **Architecture of the segmentation model.**

| Layer number | Layer |
|---|---|
| 1 | $E(\texttt{1x1n}, [64], 0.2, 0.25)$ |
| 2 | $E(\texttt{64x0n+16x1n+4x2n}, [128], 0.4, 0.25)$ |
| 4 | $E(\texttt{128x0n+32x1n+8x2n}, [256], 0.8, 0.25)$ |
| 5 | $E(\texttt{256x0n+64x1n+16x2n}, [512], 1.6, 0.25)$ |
| 6 | $D(\texttt{512x0n+128x1n+32x2n}, [512])$ |
| 7 | $D(\texttt{256x0n+64x1n+16x2n}, [256])$ |
| 8 | $D(\texttt{128x0n+32x1n+8x2n}, [128])$ |
| 9 | $D(\texttt{64x0n+16x1n+4x2n}, [64])$ |
| 10 | $\texttt{MLP}(\texttt{64x0n+16x1n+4x2n}, [128, 64], \texttt{50x0n})$ |

Table 10: **Architecture of the classification model.**

| Layer number | Layer |
|---|---|
| 1 | $E(\texttt{0x0}, [64, 128], 0.2, 0.33)$ |
| 2 | $E(\texttt{96x0n+32x1n}, [128, 256], 0.8, 0.33)$ |
| 3 | $E(\texttt{192x0n+64x1n}, [256, 512], 1.4, 0.33)$ |
| 5 | $O(\texttt{384x0n+128x1n}, [512, 256, 128], \texttt{40x0n}, 0.5)$ |

**Hardware and train times.** The training of the equivariant *learned frames + refining frames + tensor messages* model for the normal vector regression task took 46h on a single NVIDIA A100 GPU (CPU: 2 x 32-Core Epyc 7452). The training of the data-augmented model took 19h on the same machine. The best-performing equivariant classification model (learned frames + refining frames + tensor messages) was trained for 20h on a single Quadro RTX 6000 (CPU: 2 x 32-Core Epyc 7452) and the data-augmented version for 15h. The equivariant segmentation model (learned frames + refining frames + tensor messages) was trained for 39h on a single NVIDIA A100 GPU (CPU: 2 x 32-Core Epyc 7452) and the data augmented version for 17h.

