# OpenReview forum: "Beyond Canonicalization: How Tensorial Messages Improve Equivariant Message Passing"
_ICLR.cc/2025/Conference — ICLR 2025 Poster_

### Official Review · Reviewer_aU99 · 2024-11-03

**Soundness:** 3
**Presentation:** 3
**Contribution:** 2
**Rating:** 6
**Confidence:** 3

**Summary:**

The main claim of the paper is that maintaining local equivariant tensor geometric feature in the graph network is better than first canonicalize the feature in local frame and do invariant message passing.  It also proposes a way to pass tensorial features in graph network. The proposed message passing is experimented on toy shape dataset.

**Strengths:**

- The paper has a smooth and easy-to-flow presentation into equivariance.
- Maintaining local geometric features may be useful in more general settings (but not the examples shown in the paper, see weakness below). Which in the long run may benefit the equivariance community.

**Weaknesses:**

- The main concern is that all the experiments are conducted on rigid objects. However, the reviewer believes that the main advantage of the local geometric features preserving throughout the network is to deal with some non-rigid, multi-body, or deformable objects. Indeed there is no strict equivariance in deformation but it is where the local feature should make a difference. Just as shown in Fig.1 in the paper, the geometric feature should help recognize the pattern of the sub-part when it deforms or move. However, the main experiment is conducted on the modelnet rigid object, which we know the performance is quite saturated, and the reviewer believes that a robust global PCA plus any modern large point network will outperform an equivariant network in such an easy setting.
- Again, the comparison does not capture the full equivariant network baselines. We know that there are many more equivariant point networks compared on the same benchmark but they are not included in the table.
- Some more clear discussion of the difference between the proposed message passing with previous ones like TFN or VNN should be highlighted in the paper.

**Questions:**

The main concern is that the experiments are not convincing for the main claim, some more challenging cases (e.g. multi body objects)  should be included to show the effectiveness of the local geo features

---

> ### Author Response · Authors · 2024-11-20
>
> We thank the reviewer for the constructive feedback.
>
> &nbsp;
>
> **Q1/W1 (Supporting the main claim experimentally)**
> Our main claim is the following: tensorial message passing enhances performance in local canonicalization-based message passing frameworks. In the paper, we introduce tensorial message passing (Sec. 4.2) and explain that our formalism is a strict generalization, an extension, to standard local canonicalization based MPNNs (Fig. 1). In practice, we verify in several experiments on several datasets that indeed tensorial messages (in which we combine tensorial with scalar features) outperform MPNNs that use purely scalar messages (see Tab. 1, Tab. 2 and Tab. 5).
> Beyond that, Tab. 3 shows that our proposed framework substantially outperforms MPNNs with scalar messages, even when the local frames are chosen completely randomly. This highlights the key advantage of our approach: in many geometric tasks, it is indeed beneficial to enable direct communication of tensorial properties directly during message passing.
> Therefore, we firmly believe that we provide substantial experimental evidence for our main claim.
>
> We agree that there are indeed more challenging tasks that could further highlight the strengths of our method. To address this, we have conducted a number of experiments on the ScanObjectNN dataset [3], please take a look at the respective overall comment.
>
> **W2 (Related equivariant approaches)**
> Thanks for pointing out that related works are missing in the comparison tables. Since we mention TFN [1] and VNN [2] in the related work, we will add them to the comparison. If the reviewer has more suggestions, we are open to incorporating more models into our comparison.
>
> **W3 (Relation to TFN [1] and VNN [2])**
> We cite the papers TFN [1] and VNN [2] in the paragraph "Equivariance using tensorial internal representations" in the related works section. For convenience, we repeat the differences to our approach here:
>
> - Tensor Field Networks: TFN maintains equivariance throughout the network by employing equivariant layers, such as convolutions based on the tensor product. These layers rely on specialized components, including specific non-linearities or norm layers, to ensure equivariance is preserved. In contrast, our approach, like other methods based on canonicalization (see "Equivariance by canonicalization"), estimates a canonical orientation of the input features, in doing so avoiding architectural constraints imposed by specialized layers. This allows to integrate our framework more easily with existing architectures, in order to make non-equivariant architectures equivariant.
>
> - Vector Neuron Networks: VNN uses vectors within the model, ensuring that these vectors are processed in a way that preserves equivariance. Unlike TFN, VNN does not rely on the tensor product but still requires specialized layers to maintain equivariance. While VNNs are restricted to internal vector representations, our architecture imposes no such limitations, allowing for greater flexibility in the choice of representations.
>
> &nbsp;
>
> [1] Thomas et al. "Tensor field networks: Rotation- and translation-equivariant neural networks for 3D point clouds", 2018
>
> [2] Deng et al. "Vector Neurons: A General Framework for SO(3)-Equivariant Networks", Proceedings of the IEEE/CVF International Conference on Computer Vision (ICCV), 2021
>
> [3] Uy et al. "Revisiting Point Cloud Classification: A New Benchmark Dataset and
> Classification Model on Real-World Data", IEEE/CVF International Conference on Computer Vision (ICCV), 2019

---

> > ### Author Response · Authors · 2024-11-22
> >
> > We have incorporated the changes proposed by the reviewer into the paper. Please take a look at the revised version of the paper.
> >
> > We sincerely thank the reviewer for the valuable feedback. If our responses adequately address all the concerns raised, we kindly hope the reviewer will consider raising the score of our paper.

---

> ### Comment · Reviewer_aU99 · 2024-11-27
> **The authors put a great effort in responding, I raised my score from 5 to 6**
>
> Thanks to the authors for their efforts in responding to all the reviewers.
>
> Some more experiments are added to the revised version. From a theoretical perspective, I think this paper provides enough content to the community. But from a practical perspective, the problem this paper is dealing with is still a little bit toy and fake, but I believe much progress in real-world problem solving is inspired by theoretical progress. So I raised my score from 5 to 6.

---

> > ### Author Response · Authors · 2024-12-02
> >
> > Thank you for your response. We really appreciate your endorsement.

---

### Official Review · Reviewer_NqWe · 2024-11-04

**Soundness:** 4
**Presentation:** 3
**Contribution:** 3
**Rating:** 8
**Confidence:** 4

**Summary:**

This work proposes an extension of unconstrained message-passing architectures that makes them equivariant by canonicalizing the messages received by each node to its local frame.  It enables local canonicalization of arbitrary types of tensorial messages, which extends previous works that restrict the allowed tensor type of messages (e.g. only allowing for scalars or vectors).  In addition to the local canonicalization methodology, the authors introduce a mechanism for learning the local frame of each node, which is then refined in the later layers of the network.  In the experimental section, the authors evaluate their proposed method on various rotational equivariant point-cloud tasks and provide ablation studies that showcase how the individual parts of the proposed framework affect its performance and generalization.

**Strengths:**

- The authors describe the proposed framework in detail, providing clear intuition about the specific problems each part of the framework addresses.
- The simplicity of the proposed framework allows it to be easily applied to widely used non-equivariant message-passing architectures with minimal modifications.
- The experimental results demonstrate how the proposed local canonicalization benefits the performance of the baseline model when it is used in tasks with various types of tensorial outputs (e.g.  normal regression or point cloud segmentation).

**Weaknesses:**

- The proposed canonicalization procedure assumes that the inferred output vectors $v_{1},v_{2}$  are non-zero. While the authors describe how they resolve ambiguities when $v_{1},v_{2}$ are parallel, they do not explain how they handle the case where the vectors are close to zero, which makes frame selection highly sensitive to small perturbations due to noise.
-  While in Section 2 the authors mention previous works on local-canonicalization during message passing, they do not discuss work on gauge equivariant neural networks, such as the work:
 [1] Pim De Haan, Maurice Weiler, Taco Cohen, Max Welling, "Gauge Equivariant Mesh CNNs: Anisotropic convolutions on geometric graphs" ICLR (2021)
which also transforms geometric features from one local frame to another during the message passing, performed in their case during the mesh convolution.

**Questions:**

- How does the proposed method handle cases where the predicted vectors $v_{1},v_{2}$ are zero or close to zero? Additionally, how sensitive is the frame selection mechanism when different levels of noise are added to the input point clouds? Does this sensitivity change in cases of more symmetric objects?
- An addition of a discussion of gauge equivariant neural networks will benefit the completeness of the related section of this work.

---

> ### Author Response · Authors · 2024-11-20
>
> We thank the reviewer for the constructive feedback.
>
> &nbsp;
>
> **Q1.1 (Local frame estimation with predicted vectors $v_1,v_2$ (close to) zero, cf. Eq. 4)**
>
> In practice, the prediction of vectors $v_1,v_2$ close or equal to zero is extremely rare and does not cause practical instabilities. However, we identified two primary cases in which this situation could arise:
>
> 1. Nodes with no neighbors: However, if a node has no neighbors, it does not participate in message passing (given the same radius cutoff), rendering the local frame for such a node irrelevant.
> 2. Highly symmetric neighborhoods: For nodes within a highly symmetric neighborhood, establishing a meaningful local frame becomes infeasible due to the lack of distinguished directions.
>
> To address these situations, we considered two options: either one sets the local frame to zero, excluding the node from tensorial message passing or one replaces the (almost) zero vector with a random vector. We chose the latter option, as it forces the model to learn to disregard the direction of degenerate coordinate axes when no distinguished direction can be inferred.
>
> **Q1.2 (Robustness of the local frame prediction)**
>
> We are currently investigating the robustness of the estimated local frames and of the full models to noise by adding jitter to the positions of the input point clouds. We will present the results within the next days.
>
>
> **Q2 (Relation to gauge equivariant networks)**
>
> Gauge equivariant neural networks, such as the method described in [1], share a similar philosophy with our approach, including parallel transport ("frame-to-frame transitions" in our framework, cf. Fig. 1). However, key differences are:
>
> 1. Dimensionality: The method in [1] is specifically tailored for 2D meshes, whereas our architecture generalizes to arbitrary-dimensional Euclidean spaces.
> 2. Treatment of local frames: In [1], random local frames are chosen for each node, and the model is designed to be equivariant to these frames, ensuring that outputs are invariant to the specific choice of frame. In contrast, our method explicitly leverages meaningful local frames, making the model dependent on the chosen frames. This dependency allows our architecture to utilize the geometric information embedded in these frames, enhancing its expressive power in certain applications.
> 3. Constrained model architecture: The equivariance to local frames in [1] is achieved through specialized kernel constraints and specialized non-linearities, while our method imposes no restriction on the used architecture. Our approach thus offers access to a much larger design space.
>
> Furthermore, one may draw a connection to one of our ablation studies: in Tab. 3 we present results for our formalism paired with purely random local frames. In this case, the network is forced to predict (approximately) the same outputs irrespective of the choice of local coordinate frames, which can be interpreted as a form of learned gauge invariance.
>
> &nbsp;
>
> [1] Pim De Haan, Maurice Weiler, Taco Cohen, Max Welling, "Gauge Equivariant Mesh CNNs: Anisotropic convolutions on geometric graphs" ICLR, 2021

---

> > ### Author Response · Authors · 2024-11-22
> >
> > **Q1.2 (Robustness to noise)**
> > We have added a thorough discussion about the robustness of the full models and the local frame estimation in the revised version of the paper. Please take a look at the summary of revisions above and the paragraph starting at l. 909 and Fig. 5 for details.
> >
> > We sincerely thank the reviewer for the valuable feedback. If our responses adequately address all the concerns raised, we kindly hope the reviewer will consider raising the score of our paper.

---

> > > ### Comment · Reviewer_NqWe · 2024-11-27
> > > **Response to Authors' rebuttal**
> > >
> > > I thank the authors for addressing my concerns regarding the discussion of related work and for conducting additional experiments to evaluate the robustness of their proposed frame estimation module. After reading the other reviewers' comments and the authors' responses, I have decided to raise my overall score.

---

> > > > ### Author Response · Authors · 2024-12-02
> > > >
> > > > Thank you for your response. We really appreciate your endorsement.

---

### Official Review · Reviewer_xPaF · 2024-11-07

**Soundness:** 3
**Presentation:** 3
**Contribution:** 2
**Rating:** 5
**Confidence:** 5

**Summary:**

This paper focuses on the equivariant message passing and proposes a formalism which together with local canonicalization enables consistent communication of geometric features between different nodes. This method solves the problem of communicating geometric information between local patches with different coordinate frames and can be combined with other point cloud methods, achieving state-of-the-art results in the experiments.

**Strengths:**

1. The paper is well-written and technically sound.
2. The paper provides comprehensive theoretical analysis.
3. The experiment results are promising.

**Weaknesses:**

1. The proposed method relies on point normals to establish local reference frames. However, estimating accurate normals is difficult for *real-world* point clouds due to severe noise. So I expect to see results on real-world tasks rather than only synthetic datasets.
2. An important application of invariance and equivariance is point cloud registration. I expect to see the effectiveness of the proposed method on real-world point cloud registration tasks, such as 3DMatch and 3DLoMatch.
3. In Tab.3, the tensor messages surprisingly outperforms the model with scalar messages under random local frames. The random local frames affect the performance of the model in the form of noise, but these noises help tensor messages perform better. I am very curious about its reason.
4. In Tab.2, I notice refining frames brings marginal improvements, which may indicate that this step fails to obtain better normals. For comparison, I expect to see the results with (1) PCA-based normals and (2) ground-truth normals.

**Questions:**

Please address the problems in the weaknesses.

---

> ### Author Response · Authors · 2024-11-20
>
> We thank the reviewer for the constructive feedback.
>
> &nbsp;
>
> **W1 + W4 (Local frames estimation does not require point normals + PCA-based local frames)**
> We suspect a misunderstanding.
> To clarify, the construction of local reference frames in our approach does not depend on point normals. Instead, the vectors used to construct the orthonormal frames are predicted as a learned weighted sum of the relative neighbor positions (see Eq. 4).
>
> If the reviewer finds that an additional explanation in a certain section of the paper could help to prevent a potential misunderstanding, we would gladly refine the text accordingly.
>
> We thank the reviewer for the instructive suggestion to compare against local frames given by local PCA. We have re-trained our best-performing model on classification and on normal vector regression, and have replaced the learned local frames with local frames computed through local PCA (same cutoff radius as for the learned frames).
>
> We find that for the classification task, PCA-based local frames even yield a slight improvement over the learned frames (accuracy: 88.9 vs 88.7) while for normal regression the learned and iteratively refined local frames perform better (cosine similarity: 0.87 vs 0.88).
>
> Generally, we would like to highlight the flexibility in the choice of local frames as a strength of our formalism. Depending on the task and the structure of the input, different equivariant local frames can be superior and our framework can be readily applied to all equivariant estimators of rule-based or learned local frames.
>
> **W1 (Robustness to noise and real-world datasets)**
> We are currently investigating the robustness of the estimated local frames and the full models to noise in the form of adding jitter to the positions of the input point clouds. We will present the results within the next days.
> Furthermore, we conducted a number of experiments on the real-world dataset ScanObjectNN [1], for the results please see the overall comment.
>
> **W2 (Application to feature-based point cloud registration)**
> We agree with the reviewer that our framework provides an excellent formalism for building models for equivariant prediction of point features to form point correspondences for point cloud registration. We are currently working on several applications of the framework presented in the paper. We see point cloud registration as a promising avenue for future research, though setting up a new pipeline and conducting thorough experiments lies beyond the scope of this rebuttal.
>
> **W3 (Tensorial messages outperform scalar messages even with random frames)**
> Tab. 3 shows that our proposed framework (tensorial messages) outperforms traditional equivariant MPNNs based on local canonicalization (scalar messages), even when the local frames are chosen completely randomly. This is not a contradiction, but really highlights the key advantage of our framework: in many geometric tasks, it is indeed beneficial to enable direct communication of tensorial properties directly during message passing between local frames.
> In general, we believe that informative local frames improve model performance, but even with random local frames the preservation of tensorial features is beneficial (cf. Fig 1, the vectorial feature is preserved, irrespective of the local frames). The results in Tab. 3 show that the tensorial messages provide a larger gain compared to informative local frames. When the local frames are chosen fully at random, the network must learn to produce the same output irrespective of the local coordinate frame, which draws a connection to works on gauge equivariance (see also Answer to Q2 of reviewer NqWe).
>
> &nbsp;
>
> [1] Uy et al. "Revisiting Point Cloud Classification: A New Benchmark Dataset and
> Classification Model on Real-World Data", IEEE/CVF International Conference on Computer Vision (ICCV), 2019

---

> > ### Author Response · Authors · 2024-11-22
> >
> > **W1 (Robustness to noise)**
> > We have added a thorough discussion about the robustness of the full models and the local frame estimation in the revised version of the paper. Please take a look at the summary of revisions above and the paragraph starting at l. 909 and Fig. 5 for details.
> >
> > We sincerely thank the reviewer for the valuable feedback. If our responses adequately address all the concerns raised, we kindly hope the reviewer will consider raising the score of our paper.

---

> > > ### Comment · Reviewer_xPaF · 2024-12-03
> > >
> > > Thanks the authors for the detailed clarifications. The rebuttal has addressed most of my concerns but I still have two.
> > >
> > > 1. The proposed method shows comparable performance to the PCA-based method.
> > > 2. The robustness of the learned local reference frames is not ideal.
> > >
> > > I think these two problems could limit the practical value of this work. For this reason, I would keep my rating unchanged.

---

> > > > ### Author Response · Authors · 2024-12-03
> > > >
> > > > Thank you for your response.
> > > >
> > > > 1. The PCA-based experiment is not a separate method from the one we propose. In this experiment, we only modified how the local frames are predicted, while the core component of our method remains unchanged.
> > > > We view this flexibility as a strength rather than a limitation. While we propose a complete pipeline for local canonicalization, of which equivariant frame estimation is an important part, the novelty of our approach lies primarily in the tensorial message passing formulation. This formulation remains effective regardless of the specific method used for local frame estimation.
> > > >
> > > > 2. Indeed, the local frames utilized by our approach are not perfectly robust. In general, machine learning is susceptible to noise to some degree. However, we demonstrate that our models, even when trained without noisy inputs, predict fairly robust local reference frames (first coordinate axis, with noise up to twice the distance between adjacent nodes, cf. Fig. 5). Estimation of local frames with ideal robustness is an interesting avenue for future research (orthogonal to the contributions of this work), which all methods employing local canonicalization would profit from.

---

### Author Response · Authors · 2024-11-20

We cordially thank the reviewers for the time invested in reviewing our paper and appreciate their helpful comments.

&nbsp;

**Real-world dataset (W1 xPaF, W1 aU99)**
Reviewer xPaF expressed interest in the applicability to real-world datasets and reviewer aU99 suspects that for (more challenging) real-world datasets our formalism could even demonstrate larger performance improvements compared to the baseline models. To provide experimental evidence that our framework applies favorably also to real-world data, we have now conducted a series of experiments on the ScanObjectNN dataset [1] (model variants and evaluation setup as in the paper, cf. Tab 1). The dataset contains scanned indoor scene data subject to realistic measurement noise and includes deformable objects (e.g. the bag class) and multi-body objects (e.g. a shelf with objects in it). We find our method is consistently better in the $\mathrm{O}(3)/\mathrm{O}(3)$ setting than data augmentation. Furthermore, tensorial messages outperform the scalar messages by a margin, showing that our main claim is also true for noisy real-world datasets.

&nbsp;

ScanObjectNN classification accuracies:

| Method                                                    | z/z  | z/O(3) | O(3)/O(3) |
|-----------------------------------------------------------|:----:|:------:|:---------:|
| Data augmentation                                         | 87.0 | 13.3   | 70.3      |
| Learned frames + scalar messages (ours)                   | 71.9 | 71.9   | 71.9      |
| Learned frames + refining frames + scalar messages (ours) | 73.7 | 73.7   | 73.7      |
| Learned frames + tensor messages (ours)                   | 79.8 | 79.8   | 79.8      |
| Learned frames + refining frames + tensor messages (ours) | 81.0 | 81.0   | 81.0      |

$z/z$ - trained and evaluated with augmentations around the gravitational axis
$z/\mathrm{O}(3)$ - trained only with rotations around z but evaluated using all O(3) transformations
$\mathrm{O}(3)/\mathrm{O}(3)$ - trained and evaluated with augmentations all O(3) transformations

&nbsp;

[1] Uy et al. "Revisiting Point Cloud Classification: A New Benchmark Dataset and
Classification Model on Real-World Data", IEEE/CVF International Conference on Computer Vision (ICCV), 2019

---

### Author Response · Authors · 2024-11-22
**Summary of Revisions**

Once again, we would like to thank all reviewers for their time and thoughtful feedback. We greatly appreciate their constructive comments, which have helped improve the quality of our paper.

In response to all of your helpful comments, we have conducted the following experiments and added them to the paper:
- A series of experiments on the real-world dataset ScanObjectNN. We could validate our main claim that tensorial message passing significantly enhances local canonicalization also on noisy, deformable and many-body data (W1 xPaF, W1 aU99).
[paragraph from l. 899 + Tab. 6]
- Experiments using PCA-based local frames. The models with PCA-based local frames performed well on all tasks, indicating that local PCA-based frames are a very reasonable choice. It further highlights that our framework can be readily applied with both rule-based or learned local frame estimators. This is crucial for the adaptation to new datasets and domains (W4 xPaF).
[paragraph from l. 958]
- Investigation of the robustness of our model and our local frame prediction. We find that the local frames and model predictions are reasonably robust to input noise and that robustness can be further improved by training with noisy samples. Together with the experiments on ScanObjectNN this underscores the real-world applicability of our method (W1 xPaF, Q1.2 NqWe).
[paragraph from l. 909 + Fig. 5]

Additionally, we made the following changes to the paper:
- we discuss the relation to gauge equivariant networks (NqWe Q2). [l. 142]
- we included TFN and VNN as competitive related works for comparison in the results tables (aU99 W2). [Tab. 2 + Tab. 4]

We have highlighted new additional paragraphs and figures in green and refer to their locations in the paper in [...].

We hope that you find these changes satisfactory. We would be happy to answer any further questions.

---

### Meta-Review · Area_Chair_XLxj · 2024-12-23

**Metareview:**

This paper suggests a way to adapt any message passing architecture to be globally $O(d)$ equivariant by choosing (in an equivariant manner) a local frame at each node, encoding the local $R^d$ feature in this local frame, performing message passing between nodes considering these local representations and frames, and finally moving information back to global frames.

The paper is well written and suggests a simple algorithm to implement on top of existing message passing architectures. Some critique about this paper was based on: 1) the fact that the method might not be robust to noise (in frame selection process); 2) comparison to relevant previous works that also provide $O(d)$ equivariant networks is missing; and 3) the tasks/dataset considered in the experimental section are a bit too synthetic to exemplify the full merit of the method. In their rebuttal, the authors responded to these points and supported their response with adequate experiments, to the satisfaction of most reviewers.

**Additional Comments On Reviewer Discussion:**

No additional comments.

---

### Decision · Program_Chairs · 2025-01-22

Accept (Poster)